# Determination of Phenolic Compounds, Procyanidins, and Antioxidant Activity in Processed *Coffea arabica* L. Leaves

**DOI:** 10.3390/foods8090389

**Published:** 2019-09-04

**Authors:** Samuchaya Ngamsuk, Tzou-Chi Huang, Jue-Liang Hsu

**Affiliations:** 1Department of Tropical Agriculture and International Cooperation, National Pingtung University of Science and Technology, 1 Shuefu Road, Neipu, Pingtung 91201, Taiwan; samuchaya.n@gmail.com; 2Department of Biological Science and Technology, National Pingtung University of Science and Technology, 1 Shuefu Road, Neipu, Pingtung 91201, Taiwan; tchuang@mail.npust.edu.tw

**Keywords:** coffee leaves, dry processing, antioxidant activity, total phenolic content, total procyanidins

## Abstract

The effects of dry processing and maturity on antioxidant activity, total phenolic content, total procyanidins, and identity of phenolic compounds in coffee leaves were evaluated. Fresh coffee leaves were tray-dried at 40 °C for 8 h before total phenolic content, total procyanidins, and antioxidant activity were analyzed. The drying process significantly (*p* < 0.05) improved the release of total phenolic content and total procyanidins compared with the fresh leaves. The results showed that the young leaves exposed to drying processes had the highest total phenolic content, total procyanidins, and DPPH radical scavenging activity. Therefore, the effect of different drying temperatures (30, 40, and 50 °C) in the young leaves were further analyzed. The results indicated that DPPH radical scavenging activity, total phenolic content, and total procyanidins were increasingly generated when exposed to an increase in drying temperatures, whereby the highest bioactivity was evident at 50 °C. The DPPH radical scavenging activity of the coffee leaf teas was significantly correlated with total phenolic content and total procyanidins. Identification of *Coffea arabica* L. bioactive compounds by LC-MS showed the presence of catechin or epicatechin, mangiferin or isomangiferin, procyanidin B, caffeoylquinic acids (CQA), caffeine, quercetin-3-*O*-glucoside, procyanidin C, rutin, and 3,4-diCQA. *Coffea arabica* L. leaf tea was confirmed to be a potential functional food rich in phenolic compounds with strong antioxidant activity.

## 1. Introduction

Coffee leaves have been largely neglected and considered to have little or no value to farmers due to the high value placed on coffee beans. Nonetheless, coffee leaves have been used as traditional medicine to treat or mitigate various diseases or disorders. Recently, increased focus has been paid to coffee leaves as an alternative to traditional tea consumption due to the fact or their high levels of polyphenolic compounds and great potential as a healthy beverage [1]. For example, in countries like Indonesia, Jamaica, India, Java, Sumatra, Ethiopia, and South Sudan, sun-dried coffee leaves have been used as a traditional tea substitute [2]. Coffee leaf tea has higher antioxidant potential and lower caffeine than traditional tea. The polyphenol content of coffee leaf depends on the maturity, the harvest times (month of harvest), and the coffee species of its leaf and are phytochemically composed of caffeine, trigonelline, adenine-7-glucosyl, theobromine, theophylline, *ent*-kaurane diterpenpids, 7-methylxanthine, anthocyanins, mangiferin, isomangiferin, catechin, epicatechin, procyanidin B1, chlorogenic acid (5-CQA), glucoside, rutin, isorhamnetin, quercetin, isoquercitrin, kaempferol, histidine, pipecolic acid, sucrose, tannins, caffeic acid, p-coumaric acid, ferulic acid, sinapic acid, neochlorogenic acid (3-CQA), and cryptochlorogenic acid (4-CQA) [1,3,4,5,6,7,8].

Procyanidins are flavonoids under the proanthocyanidin class with potent antioxidant activity, being able to scavenge a wide range of free radical species and nitrogen species [9,10,11]. Many studies have shown the antioxidant potential of procyanidins. The antioxidant activity of procyanidins from *Ceruios tagal* leaves showed strong ferric reducing antioxidant power (FRAP) and 2,2-diphenyl-1-picrylhydrazyl (DPPH) radical scavenging activity [12]. Similarly, Chen et al. (2014) [13] reported that the procyanidins extracted from *Polyalthia longifolia* leaves had higher DPPH radical scavenging activities, 2,2′-azino-bis(3-ethylbenzothiazoline-6-sulphonic acid) (ABTS) free radical scavenging activities, and FRAP as compared with ascorbic acid.

Traditional tea processing to produce the different categories of tea can be generally distinguished by the process they undergo—non-fermented and fermented. Basically, tea leaves either involve wilting different forms and degrees of fermentation or not, ending the fermentation if any, shaping the tea and drying [14]. Alterations in food characteristics can be attributed to the changes in phenolic compounds due to presence of heat and oxidation during drying [15]. Previous studies suggest that the phytochemical properties of coffee leaf tea are affected by the type of processing method, coffee leaf maturity, and brewing time [16]. According to Yamassaki et al. (2017) [17], the levels of total phenolic content and antioxidant activity in avocado leaves could be increased with decreased drying temperature. Coffee leaves contain similar phenolic compounds to that of tea (*Camellia sinensis*) leaves, hence, traditional tea processing methods used to process coffee leaves could alter the phytochemical composition and bioactivity of the final product.

The present study, therefore, aimed to (I) identify the phenolic compounds of unfermented coffee leaf extracts after drying; (II) assess the effects of different drying temperatures on their total phenolic content, total procyanidins and antioxidant activity; and (III) evaluate their correlation.

## 2. Materials and Methods

### 2.1. Materials

Coffee leaves were collected at the campus of National Pingtung University of Science and Technology, Taiwan and the samples were divided into 2 groups based on maturity—young (young leaves were on the apex) and mature (mature leaves were the first node as below the apex to second node) leaves. Acetonitrile (ACN) and trifluoroacetic acid (TFA) were purchased from Merck KGaA (Darmstadt, Germany); methanol from Honeywell (Ulsan, Korea); butyl hydroxyl anisole (BHA), dimethyl sulphoxide (DMSO), 2,2-diphenyl-1-picrylhydrazyl (DPPH), formic acid (FA), Folin–Ciocalteu phenol reagent, gallic acid and sodium carbonate from Sigma–Aldrich (St Louis, MO, USA); and Vanillin from Panreac Química SLU (Castellar del Vallés, Barcelona, Spain), while procyanidin B1 was obtained from Sunhank (Tainan, Taiwan).

### 2.2. Processing and Extraction of Coffee Leaves

Fresh coffee leaves (young and mature) were plucked and washed by hand. The coffee leaves were sun-dried and rolled. One batch (samples were harvested in the winter season) of the rolled coffee was tray-dried for 8 h at 40 °C to analyze the effect of the drying process, and three other batches (samples were harvested in the rainy season) were tray-dried at different temperatures (30, 40 and 50 °C for 8 h), respectively, to compare the effect of drying temperatures. The dried coffee leaf samples were pulverized to a homogenous powder in a laboratory grinder. The fresh coffee leaves were cut to small pieces. Samples were labeled as fresh-young leaves, fresh-mature leaves, dried-young leaves, and dried-mature leaves after processing and grinding. Five grams of fresh samples or dried samples either at young or mature age were extracted in triplicate with 70% of methanol at room temperature for 3 h. The supernatant was filtered through a 0.22 µm syringe filter and the filtrates were stored at −20 °C for further experiments.

### 2.3. HPLC Analysis of Bioactive Compounds of Coffee Leaves

The method to determine the bioactive compounds of coffee leaf was adapted from Thiesen et al. (2016) [18]. High-performance liquid chromatography (HPLC) analysis was conducted on a Hitachi HPLC system (Hitachi, Tokyo, Japan) equipped with a C30 column (250 × 4.6 mm, 5 µm). Samples (20 µL) were injected into the system. The mobile phase was composed of 95% acetonitrile + 0.01% trifluoroacetic acid (solvent A) and 5% acetonitrile + 0.01% trifluoroacetic acid (solvent B) at 1 mL/min. The gradient program was as follows: 10% A from 0–5 min; 10–15% A from 5-7 min; 15–20% A from 7–13 min; 20–25% A from 13–20 min; 25–30% A from 20–30 min; 30–80% A from 30–35 min, and 10% A from 35–40 min then returning to initial conditions. Detection of phenolic compounds was carried out with a UV-visible detector at 278 nm. Quantification of the compounds was done using procyanidin B1 as the standard.

### 2.4. Liquid Chromatography Mass Spectrometer (LC-MS)

Liquid chromatography mass spectrometer analysis was performed on a Thermo Finnigan LCQ Deca XP Max LC/Msn (California) fitted with a C30 column (250 × 4.6 mm, 5 µm). Phenolic compounds were detected by their UV absorbance at 278 nm. Twenty microliters of samples were injected into the system. The mobile phase, composed of a mixture of Solvent A (95% acetonitrile + 0.01% formic acid) and Solvent B (5% acetonitrile + 0.01 formic acid), was set at 1 mL/min. The gradient program was identical to that of the analytical HPLC for ESI source, the inlet conditions were capillary voltage 20 V; temperature 300 °C; sheath gas flow rate 50 arb; spray voltage 4 kV, and the mass spectrometer was operated on negative ion mode. The ion trap mass spectrometer was operated in the m/z (‒) 100–1000 for the MS scan. The data were analyzed with the ThermoXcaliburTM computer program.

### 2.5. Measurement of DPPH Radical Scavenging Activity

The method to determine the scavenging activity of coffee leaf extracts was adapted Chao et al. (2010) [19]. A DPPH solution of 1 mM was prepared in 99.5% methanol. One milliliter samples (10 mg/mL) or Butyl hydroxyl anisole (BHA) standards (50 ppm) were mixed with 250 µL of DPPH solution and incubated for 30 min in the dark at room temperature. Absorbance was then measured at 517 nm in a SpectraMax 190 Microplate Reader (Molecular Devices, LLC, San Jose, CA, USA) after 200 µL of the mixture were added into 96 well plates in triplicate. Methanol was used as the blank solution.

The percentage inhibition of the DPPH free radical was calculated as:% Inhibition = (Abs_blank_ − Abs_sample_)/(Abs_blank_) × 100.(1)
where, Abs_sample_ = absorbance of 1 mmol DPPH with sample in methanol and Abs_blank_ = absorbance of methanol solvent in absence of DPPH and sample.

### 2.6. Measurement of Total Phenolic Content

Total phenolic content in the coffee leaf extracts was determined by a modified method of Miliauska et al. (2004) [20]. One hundred microliter of gallic acid standards or sample solutions (10 mg/mL) were mixed with 500 µL of 10 fold diluted Folin–Ciocalteu phenol reagent and 400 µL of 0.7 M sodium carbonate, and incubated at room temperature in the dark for 30 min. Two hundred microliter of the solution were then added into 96 wells plates and absorbance at 765 nm was determined using a SpectraMax 190 Microplate Reader (Molecular Devices, LLC, San Jose, CA, USA). Total phenolic content was expressed as mg gallic acid/g of leaf using the equation obtained from a calibration curve of gallic acid. All samples were measured in triplicate.

### 2.7. Measurement of Total Procyanidins

Total procyanidins were estimated using the method developed by Sun et al. (1998) [21]. A mixture of 3 mL 4% (*v*/*v*) Vanillin–methanol solution and 1.5 mL of hydrochloric acid was prepared beforehand. Point five microliter of sample extracts (10 mg/mL) or gallic acid standards were then added to the mixture and vortex at room temperature for 15 min before absorbance was read at 500 nm using a SpectraMax 190 Microplate Reader (Molecular Devices, LLC, San Jose, CA, USA). Total procyanidins was expressed as mg gallic acid/g of leaf using the equation obtained from calibration curve of gallic acid.

### 2.8. Statistical Analysis

Experimental results were used for analysis of variance using SPSS version 17 (IBM, Armon, NY, USA). Significant differences between means were determined by Duncan’s multiple-range test (*p* < 0.05). Principal component analysis (PCA) was used in this study to establish the relationships among the different variables.

## 3. Results

### 3.1. Effect of Drying Process on DPPH Radical Scavenging Activity

The influence of maturity and drying on the antioxidant activity of coffee leaf extracts was measured by DPPH radical scavenging activity (Table 1). Total antioxidant activity was significantly associated with drying and maturity (*p* < 0.05). According to the scavenging activity results, the highest activity was found in dried, young leaf extracts (95.01 ± 0.44%) dried for 8 h at 40 °C. The results from this assay were similar to those of Rabeta and Lai (2013) [14], in which the unfermented tea samples had higher DPPH radical scavenging activity than fermented tea. Moreover, young coffee leaves processed with a Japanese-style green tea method had the lowest IC_50_ for DPPH radical scavenging activity [16]. Similarly, mulberry leaves dried at 40 °C exhibited the best and strongest antioxidant activity (89.48 ± 37.65 ppm) [22]. According to Qa’san et al. (2011) [23] dried *Ginkgo biloba* leaf flavan-3-ols extracts, as indicated by their lower IC_50_. This suggests that antioxidant activity is directly related to total phenolic content and total procyanidins, which are affected by the drying processes [24]. According to Hlahla (2010) [25], unfermented bush tea had higher antioxidant activity than fermented bush tea because its scavenging activity was reduced after processing, unlike unfermented bush tea in which it was produced after processing. Many studies indicated that the leaves showed high antioxidant activity because of the high amount of pigments created, which in turn protects the leaf tissue phytochemicals [26,27]. Antioxidant activity has been positively correlated with the content of total phenolic compounds in leaf extracts, especially after the drying process. Hence, the intake of coffee leaf teas as an addition to the food industry can have promising health benefits [28].

### 3.2. Effect of Drying Process on Total Phenolic Content

It has been established that the biologically active compounds in plants are responsible for their antioxidant activity, hence, it is logical to measure the total phenolic content in the leaf extracts [29]. Total phenolic content as determined by the Folin–Ciocalteu colorimetric method of teas from coffee leaves exposed to a drying process is shown in Figure 1. The highest amounts were found in the extracts of young leaves dried at 40 °C for 8 h. Based on the maturity of coffee leaf extracts, young leaves had total phenolic compounds higher than those of mature leaves. Previous research likewise suggests that young coffee leaves contain higher total phenolic compounds than coffee mature leaves and old coffee leaves [16]. Based on the drying process, similar results to those of this study were reported by Cheng et al. (2006) [30] who expressed that the high total phenolic content in dried samples may occur due to the increase in the number of bound phenolic compounds as a part of the breakdown of cellular constituents during the drying process. Similarly, Bhakta and Ganjewala (2009) [31] reported that the dried, young leaves of *Lantana camara* L. had higher total phenolic compounds than dried mature and old leaves. According to Haddadi et al. (2018) [32], the leaves of green papaya contained higher total phenolic compounds than skin and pulp. Rebaya et al. (2016) [33] also reported that higher contents of total phenolic compounds were evident in rockrose (*Cistus salviifolius*) leaves compared to its flowers. Another study also found higher total phenolic compounds in *Viscum album* leaves extracts compared with twigs [34]. The increase in total phenolic content is related to the association of phenolic compounds with their available precursors arising from non-enzymatic inter conversion between the phenolic molecules [35]. Phenolic compounds could be released from collapsed cell walls into the solvent and drying treatments helps to release large amounts of bound phytochemicals into the medium, namely, phenolic compounds from the matrix, thus making them more accessible by extraction solvent as well [36]. This is probably the reason that low total phenolic compounds, DPPH radical scavenging activity and total procyanidins were evident in fresh coffee leaves.

### 3.3. Effect of Drying Process on Total Procyanidins

Total procyanidins as affected by drying process were determined in coffee leaf extracts as shown in Figure 1. The procyanidins content was expressed in gallic acid equivalents. The highest total procyanidins content was found in extracts of young coffee leaves exposed to drying at 40 °C for 8 h. Procyanidins are important compounds in terms of antioxidant properties [9,10,11]. Bhakta and Ganjuewala (2009) [33] reported that dried young leaves of *Lantana camara* L. had higher total procyanidins than dried mature and old leaves. According to Navarro-Hoyos et al. (2017) [37] total procyanidins were higher in *Unaria tomentos* L. leaves than in its bark, stems, and wood. Ramsay and Mueller-Harvet (2016) [38] also reported that dried *Averrhoa bilimibi* leaves contained higher procyanidins than its fruit, 4.5 g/100 g dry weight and 2.2 g/100 g dry weight, respectively. Phytosterols are found in either young, mature or old leaves. However, the quantity of total phenolic compounds and procyanidins in leaves is greatly dependent on the maturity of leaves [39].

### 3.4. Effect of Different Drying Temperatures on DPPH Radical Scavenging Activity, Total Phenolic Content, and Total Procyanidins

Based on the results of the effect of drying process, this study found that dried, young coffee leaves had the highest DPPH radical scavenging activity, total phenolic content, and total procyanidins content. Therefore, the effect of three different drying temperatures (30, 40, and 50 °C) on DPPH radical scavenging activity, total phenolic content, and total procyanidins in young *Coffea arabica* leaves was further studied. Figure 2A shows the DPPH radical scavenging activity after drying young coffee leaves at different temperatures. The DPPH radical scavenging activity of dried, young coffee leaves was evidently increased with increasing drying temperature. The young leaves dried at 50 °C exhibited the highest radical scavenging activity. Furthermore, the drying temperature also caused an increase in total phenolic content and total procyanidins (Figure 2B). Total phenolic content and total procyanidins were the highest in young leaves dried at 50 °C. Similar results reported by Katsube et al. (2009) [40] indicated that DPPH radical scavenging activity and total phenolic compounds increased when the drying temperature of mulberry leaves was increased from 40–60 °C. According to Yamassaki et al. (2017) [17], the drying temperatures of 40, 50, and 60 °C applied to avocado leaves showed an increase in total phenolic content and DPPH radical scavenging activity when compared to those that had been dried at 70, 90, and 100 °C. In addition, total phenolic content increased when the drying temperature increased (40, 50, 60, 70, and 80 °C) in *Borassus aethiopum* mart ripe fruit pulp [41]. Moreover, dried *Stevia rebaudiana* (Bertoni) leaves showed increased total phenolic compounds with temperature increase according to Ciulu et al. (2017) [42]. Furthermore, Figueroa et al. (2018) [43] indicated that procyanidins in dried avocado peel seemed to be the phenolic compounds with the content that varied the most with temperature increase.

### 3.5. Identification of Phenolic Compounds in Leaf Extracts of Coffea arabica *L.*

High Performance Liquid Chromatography was used to separate major phenolic compounds in *Coffea arabica* L.-processed extracts and detected with UV at 278 nm. The identification of phenolic compounds was performed and confirmed by respective high-resolution mass-to-charge (m/z) from LC-MS. All analytical data for the examined compounds are listed in Table 2. Nine active compounds in the fresh and processed sample extracts were detected by HPLC after 40 min. Among these compounds, three procyanidins were identified. Catechin or epicatechin, procyanidin B, and procyanidin C were detected at HPLC retention times of 2.17, 5.92, and 12.92 min, respectively (Table 2). Moreover, compounds were identified as [M-H]^−^ quasi-molecular ions by negative ESI-MS using LC-MS with the same conditions as that of HPLC (Figure 3A–C). Results from LC-MS showed mass spectrum of [M-H]^−^ m/z 289 (290), 577, and 865 for catechin or epicatechin, procyanidin B, and procyanidin C, respectively. The mass spectrometer used in this study was a unit-resolution mass spectrometer; therefore, the meaningless decimal parts were in column of the [M-H]^−^. Murakami et al. (2006) [44] reported that oolong bottled tea drinks had mass spectrums of [M-H]^−^ m/z 289 (catechin) and 577 (procyanidin B1 and procyanidin B2) after LC-MS analysis. These results were also supported by Fraser et al. (2012) [45] with mass spectrums of m/z 289 for monomer procyanidin (catechin or epicatechin), 577 for dimer procyanidin, and 869 for trimer procyanidin in grape seed, green tea, oolong tea, and black tea. Similarly, Falleh et al. (2011) [46] identified three kinds of procyanidins compounds in *Mesembryanthemun edule* L. with mass spectra of [M-H]^−^ m/z 289, 577, and 865 indicative of (‒) epicatechin, procyanidin B, and procyanidin trimer, respectively. To elucidate the major compounds in the dried coffee leaves that are responsible for the antioxidant activity, active compounds in the MeOH extract from coffee leaves dried at 40 °C was prepared and subjected to LC-MS analysis. Phenolic compounds were identified by comparing with reference standard or according to their MSn and MS/MS spectra. The MS/MS spectra of m/z 289 (290), 577, and 865 are shown in Figure 3D–F, respectively. The MS/MS spectrum of m/z 577.13 (at retention times 5.92 min) in the negative ion mode show product ions at m/z 451, 425, 407, 289, and 287 which were consistent with those of proanthocyanidin dimers previously reported in grape seed [21]. The MS/MS spectrum of m/z 865.5 (at retention times 12.94 min) in the negative ion mode showed product ions at m/z 739, 713, 695, 577, 575, 499, and 289, which are consistent with those of procyanidin trimers previously reported in grape seed [47].

### 3.6. Effect of Drying Process on Phenolic Compounds in Coffea arabica *L*.

According to the preliminary results, processing methods and maturity of coffee leaves affect the quantity of phenolic compounds as shown in Table 2. The procyanidins polymerization group, i.e., procyanidin B in fresh, young leaves; fresh, mature leaves; and dried, mature leaves were relatively lower than in dried, young leaves. Moreover, procyanidin C in dried mature leaves was evidently higher than in the other sample, while, catechin was not significantly different among all samples. Our study found that the fresh, young leaves; fresh, mature leaves; dried, young leaves; and dried, mature leaves contained mangiferin or isomangiferin at 20.34, 19.03, 49.84, and 42.46 mg/g, respectively; procyanidin B at 17.40, 12.49, 25.40, and 18.41 mg/g, respectively; caffeoylquinic acid (CQA) at 14.00, 12.01, 19.30, and 15.17 mg/g, respectively; and caffeine at 79.79, 80.65, 320.08, and 259.31 mg/g, respectively. Moreover, quercetin-3-*O*-glucoside was also found in fresh, young leaf (16.34 mg/g); fresh, mature leaf (11.99 mg/g); dried, young leaf (22.65 mg/g); and dried, mature leaf (17.27 mg/g) samples. Furthermore, the procyanidin C, rutin, and 3,4 dicaffeoylquinic acid contents were 88.18, 23.76, and 11.37 mg/g, respectively, in fresh, young leaves; 88.12, 12.34, and 11.39 mg/g; respectively, in fresh, mature leaves; 121.76, 24.24, and 12.00 mg/g, respectively, in dried, young leaves; and 148.92, 14.84, and 11.39 mg/g, respectively, in dried, mature leaves. Similar results for epicatechin, mangiferin isomangiferin, procyanidin B, rutin, and quercetin glucoside derivatives contents were reported in coffee leaves (Almeida et al., 2014) [7]. In addition, coffee leaf samples in this study contained seven (i.e., caffeine, quercetin glucoside, rutin, mangiferin, isomangiferin, catechin, and epicatechin) bioactive compounds reported in commercial coffee leaves [1,4,5,6].

### 3.7. Effect of Different Drying Temperatures on Phenolic Compounds in Coffea arabica *L*.

In order to investigate the effect of different drying temperatures on the quantity of the bioactive compounds found in young coffee leaves after preliminary results indicated that young, dried leaf samples had higher total phenolic content, young coffee leaf samples were dried at 30 °C, 40 °C, and 50 °C (Table 3). Catechin or epicatechin content was not significantly affected by the temperature difference when compared with fresh, young leaf samples. However, all the other compounds were significantly increased with the increase in temperature. Mangiferin or isomangiferin was found in young leaf samples dried at 30 °C (39.26 mg/g), at 40 °C (86.09 mg/g), and at 50 °C (108.78 mg/g). Moreover, young leaf samples dried at 30 °C, 40 °C, and 50 °C contained procyanidins B at 29.86, 41.16, and 57.79 mg/g, respectively; caffeoylquinic acids at 15.31, 18.17, and 20.23 mg/g, respectively; and caffeine at 160.83, 301.66, and 361.93 mg/g, respectively. Furthermore, quercetin-3-glucoside was found in young leaves dried at 30 °C (16.56 mg/g), at 40 °C (20.13 mg/g), and at 50 °C (25.94 mg/g). For procyanidin C was found in young leaf samples dried at 30, 40, and 50 °C contained 115.26, 178.16, and 351.07 mg/g, respectively. Rutin was found in young leaf samples dried at 30 °C (25.15 mg/g), at 40 °C (28.69 mg/g), and at 50 °C (39.75 mg/g). Finally, the mean content of 3,4-dicaffeoylquinic acid in young coffee leaves dried at 30 °C, 40 °C, and 50 °C were 16.01, 16.35, and 18.82 mg/g, respectively. The phenolic compounds in *Coffea arabica* L. were increasingly generated when exposed to an increase in drying temperatures.

### 3.8. Correlation of Drying Process on Antioxidant Activity, Total Phenolic Content, and Total Procyanidins

Principle component analysis (PCA) was used to establish the relationship between the antioxidant activity (DPPH), total phenolic content, and total procyanidins on the four extracts from coffee leaves (i.e., fresh young, fresh mature, dried young, dried mature) (Figure 4). The two principle components were separated into component 1 and component 2 of the total variance in the dataset. The principle component analysis 1 showed high correlation between total phenolic compounds and DPPH radical scavenging. In addition, DPPH radical scavenging and total phenolic compounds were significantly correlated with each sample. The principle component analysis 2 was positively correlated with total procyanidins.

## 4. Conclusions

Nowadays, extensive research is being carried out to find functional foods from natural products. Food processing can significantly influence the phytochemical and bioactive activity of foods. This study illustrated the effect of drying processes on phenolic contents in coffee leaves. The different stages and different temperatures caused significant differences in the antioxidant activity, total phenolic content, and total procyanidins. The bioactive compounds, DPPH radical scavenging activity, total phenolic content, and total procyanidins increased when the drying temperature increased to 50 °C. Moreover, high correlation was evident between total phenolic content and total procyanidins and between total phenolics and DPPH radical scavenging activity.

## Figures and Tables

**Figure 1 foods-08-00389-f001:**
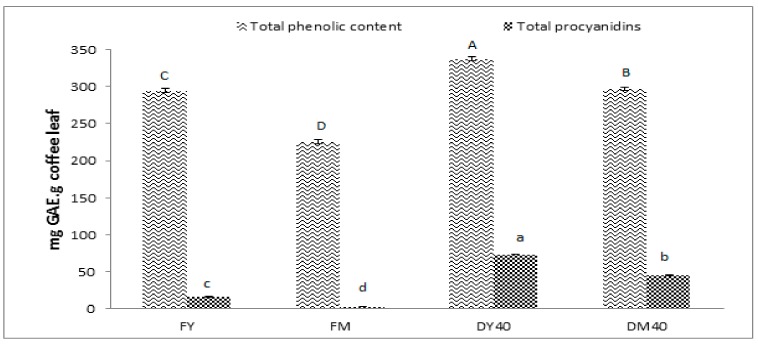
Total phenolic content and total procyanidins in coffee leaf teas, fresh young (FY); fresh mature (FM); dried young 40 °C, 8 h (DY40); dried mature 40 °C, 8 h (DM40), compared with gallic acid (*p* < 0.05).

**Figure 2 foods-08-00389-f002:**
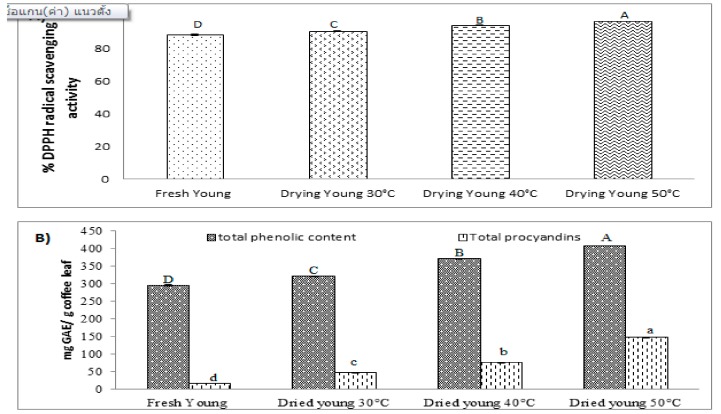
(**A**) DPPH radical scavenging activity. (**B**) Total phenolic content and total procyanidins in fresh young, dried young 30 °C, dried young 40 °C, and dried young 50 °C coffee leaves. Different letters indicate the significance in each group with other treatments (*p* < 0.05).

**Figure 3 foods-08-00389-f003:**
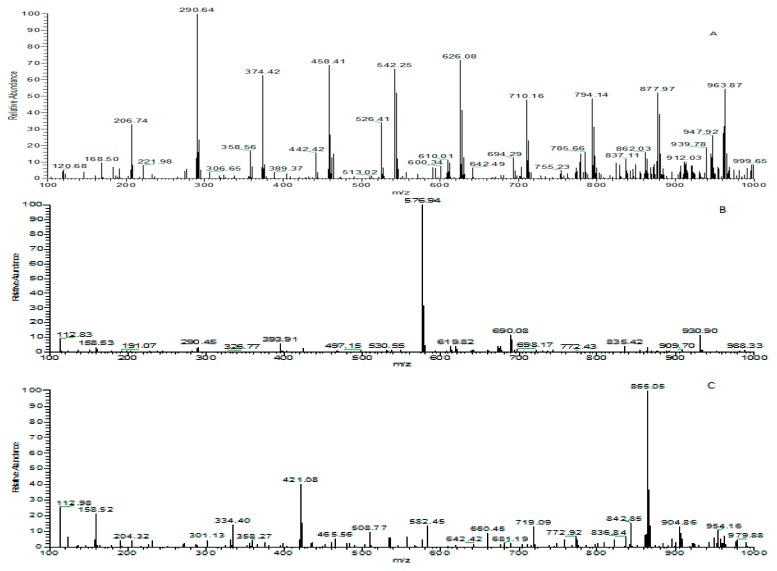
MS and MS/MS spectra of extracts isolated from dried, young coffee leaves. (**A**) MS spectrum of m/z 290 (289), (**B**) MS spectrum of m/z 577, (**C**) MS spectrum of m/z 865, (**D**) MS/MS of spectrum of m/z 290 (289), (**E**) ESI MS/MS spectrum of m/z 577, and (**F**) MS/MS of spectra of m/z 865. All MS and MS/MS spectra were performed under negative mode. Due to the mass inaccuracy of mass spectrometry, m/z 289 was sometimes shifted to m/z 290.

**Figure 4 foods-08-00389-f004:**
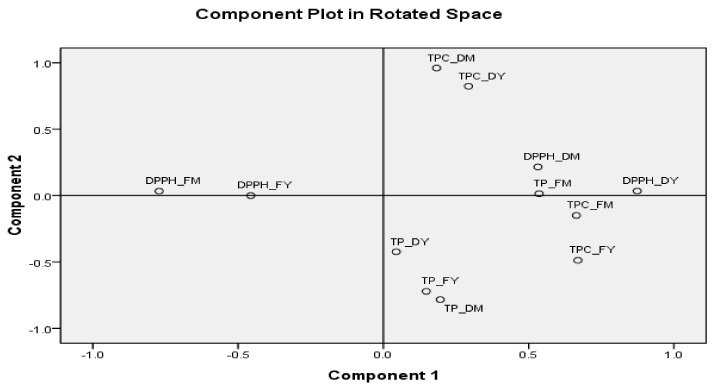
Principle component analysis of DPPH radical scavenging activity (DPPH), total phenolic content (TPC), and total procyanidins (TP) in coffee leaf teas (young (Y) and mature (M)) and processing methods (fresh (F) and drying (D)).

**Table 1 foods-08-00389-t001:** Antioxidant activity of drying process on coffee leaf tea (*Coffea arabica* L.).

Samples	%DPPH Radical Scavenging Activity
Butyl hydroxyl anisole (BHA)	91.04 ± 0.84 ^E^
Fresh Young	92.93 ± 0.51 ^C^
Fresh Mature	92.24 ± 0.95 ^D^
Dried Young 40 °C, 8 h	95.01 ± 0.44 ^A^
Dried Mature 40 °C, 8 h	93.40 ± 0.70 ^B^

^A,B,C,D,E^ The vakues were expressed as mean ± standard deviation from at least three individual experiments. Different letters indicate the significant difference in each group with other treatments (*p* < 0.05), Duncan test.

**Table 2 foods-08-00389-t002:** Identification and quantification of phenolic compounds in *Coffea arabica* L.

Compound Number	Compound	Molecular Formula	RT (min)	[M-H]^−^ (m/z)	FY (mg/g)	FM (mg/g)	DY40 (mg/g)	DM40 (mg/g)
1	Catechin or Epicatechin	C_15_H1_4_O_6_	2.17	289	11.24 ± 0.02 ^a^	11.21 ± 0.04 ^a^	11.23 ± 0.06 ^a^	11.23 ± 0.03 ^a^
2	Mangiferin or Isomangiferin	C_19_H_18_O_11_	3.33	421	20.34 ± 0.06 ^c^	19.03 ± 0.07 ^d^	49.83 ± 0.01 ^a^	42.46 ± 0.05 ^b^
3	Procyanidin B	C_30_H_26_O_12_	5.92	577	17.40 ± 0.05 ^c^	12.49 ± 0.02 ^d^	25.40 ± 0.04 ^a^	18.41 ± 0.07 ^b^
4	Caffeoylquinic acids (CQA)	C_16_H1_8_O_9_	8.37	353	14.00 ± 0.10 ^c^	12.01 ± 0.11 ^d^	19.30 ± 0.02 ^a^	15.17 ± 0.03 ^b^
5	Caffeine	C_8_H_10_N_4_O_2_	9.43	193	79.79 ± 0.02 ^d^	80.65 ± 0.03 ^c^	320.08 ± 0.07 ^a^	259.31 ± 0.07 ^b^
6	Quercetin-3-O-glucoside	C_21_H_19_O_12_	12.36	463	16.34 ± 0.15 ^c^	11.99 ± 0.18 ^d^	22.65 ± 0.09 ^a^	17.27 ± 0.02 ^b^
7	Procyanidin C	C_45_H_38_O_18_	12.94	865	88.18 ± 0.06 ^c^	88.18 ± 0.08 ^c^	121.76 ± 0.02 ^b^	148.02 ± 0.05 ^a^
8	Rutin (isomer1 or isomer2)	C_27_H_30_O_16_	18.11	609	23.76 ± 0.01 ^a^	12.34 ± 0.03 ^c^	24.24 ± 0.05 ^a^	14.84 ± 0.01 ^b^
9	3,4-Dicaffeoylquinic acid (3,4-diCQA)	C_25_H_24_O_12_	20.01	515	11.37 ± 0.09 ^b^	11.39 ± 0.05 ^b^	12.00 ± 0.06 ^a^	11.39 ± 0.07 ^b^

^a,b,c,d^ The values are expressed as mean ± standard deviation from at least three individual experiments. The compounds in *Coffea arabica* L. are from fresh young (FY), fresh mature (FM), dried young 40 °C (DY40), and dried mature 40 °C (DM40) leaf. Different letters indicate the significant difference in each group with other treatments (*p* < 0.05), Duncan’s test.

**Table 3 foods-08-00389-t003:** Analysis of bioactive compounds in *Coffea arabica* L. dried at 30, 40, and 50 °C.

Compound Number	Compound	FY (mg/g)	DY30 (mg/g)	DY40 (mg/g)	DY50 (mg/g)
1	Catechin or Epicatechin	11.24 ± 0.02 ^a^	11.21 ± 0.08 ^a^	11.23 ± 0.03 ^a^	11.25 ± 0.04 ^a^
2	Mangiferin or Isomangiferin	20.34 ± 0.06 ^d^	39.26 ± 0.05 ^c^	86.09 ± 0.02 ^b^	108.78 ± 0.04 ^a^
3	Procyanidin B	17.40 ± 0.05 ^d^	29.86 ± 0.01 ^c^	41.16 ± 0.07 ^b^	57.70 ± 0.05 ^a^
4	Caffeoylquinic acids (CQA)	14.00 ± 0.10 ^d^	15.31 ± 0.10 ^c^	18.17 ± 0.08 ^b^	20.23 ± 0.02 ^a^
5	Caffeine	79.79 ± 0.02 ^d^	160.83 ± 0.03 ^c^	301.66 ± 0.07 ^b^	361.93 ± 0.06 ^a^
6	Quercetin-3-*O*-glucoside	16.34 ± 0.15 ^c^	16.56 ± 0.04 ^c^	20.13 ± 0.02 ^b^	25.94 ± 0.05 ^a^
7	Procyanidin C	88.18 ± 0.06 ^d^	115.26 ± 0.06 ^c^	178.16 ± 0.01 ^b^	351.07 ± 0.08 ^a^
8	Rutin (isomer1 or isomer2)	23.76 ± 0.01 ^d^	25.15 ± 0.03 ^c^	28.69 ± 0.07 ^b^	39.75 ± 0.04 ^a^
9	3,4-Dicaffeoylquinic acid (3,4-diCQA)	11.37 ± 0.09 ^c^	16.01 ± 0.09 ^b^	16.35 ± 0.03 ^b^	18.82 ± 0.05 ^a^

^a,b,c,d^ The values are expressed as mean ± standard deviation from at least three individual experiments. The compounds in *Coffea arabica* L. are from fresh young (FY), dried young 30 °C (DY30), dried young 40 °C (DY40), and dried young 50 °C (DY50) leaf. Different letters indicate the significant difference in each group with other treatments (*p* < 0.05), Duncan’s test.

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
