# Peer review of "Determination of Phenolic Compounds, Procyanidins, and Antioxidant Activity in Processed Coffea arabica L. Leaves"

_foods, 2019, doi:10.3390/foods8090389_

Round 1
Reviewer 1 Report
The study is well conducted and is on a currently relevant topic; however, the manuscript has to be thoroughly revised in terms of English writing. It is evident that the authors have used several methods to identify and quantify the phenolic content along with corresponding antioxidant activity. Check the writing and English mistakes:
Page 1, line 40, remove extra stop mark “substitute [2]..” Page 2, line 54, remove “assays” and replace “when” with “as” Page 2, line 66, replace “aims” with “aimed” Page 3, line 105, add “,” after bracket “)” Line 172, add “.” after [14]Section 2.2, this section has to be explained in a better way. There are few queries, which should be dealt with.
Why were the leaves sun dried and rolled, before conducting the drying experiment. Does the author mean “sun dried OR rolled”, after which the rolled leaves were subjected to the drying analysis. Which samples were used as the standard samples for comparison, were they sun dried samples? Line 82-83, were the drying times differing, or were they all conducted for 8 hours. Also, how was it decided, when to stop drying? Line 84, how could the fresh coffee sample be pulverised to powder? Line 85, which samples were designated, which names, authors need to be clearer.Page 2, line 91, add “from the method developed by Author-name [16]”
Line 132, modify the sentence. Replace “using the method [19].” with “using the method developed by Author-name [19].”
There are several mistakes like these, where sentences need modification, along with addition of Author names.
Line 150, Line 154, citation [21] should be “Author-name [21]” Line 157, citation [23] should be “Author-name [23]” Line 175, citation [29] should be “Author-name [29]” Line 176, citation [30] should be “Author-name [30]” Line 177 and 192, citation [31] should be “Author-name [31]” Line 194, citation [35] should be “Author-name [35]” Line 195, citation [36] should be “Author-name [36]” Line 211, citation [38] should be “Author-name [36]” Line 213, citation [15] should be “Author-name [15]” Line 218, citation [40] should be “Author-name [40]”. Similarly, for [41]. Page 2, line 51, Write the Author name for the citation [11] Page 2, line 61, Write the Author name for the citation [15]Authors should search similar mistakes and modify the sections.
Rephrase the sentences
Line 154-155, “According …..”,. Line 218-220, “On the other hand….” Line 299-301, so what does it imply.
The Result and discussion section needs thorough rearrangement. Some sections seem repeated. Authors don’t need to mention the results already documented in Tables and Figures and can describe about the trend and the reason behind the observations.
Author Response
Dear reviewer
I revised the manuscript just a little bit (similar your suggestion) and add file answer
Why were the leaves sun dried and rolled, before conducting the drying experiment. Does the author mean “sun dried OR rolled”, after which the rolled leaves were subjected to the drying analysis.= The leaves were sun dried and rolled before conduction the drying to remove excess water from the leaves. In addition, sun drying could increase free amino acids, the availability of phenolic compounds as well as the change of taste. After sun drying, the leaves were passed to the rolling action that affected to some sap, essential oils and water inside the leaves to ooze out resulting in the enhancement of leaves taste.
Which samples were used as the standard samples for comparison, were they sun dried samples?
= Fresh leaves were used as the standard sample for comparison in this experiment.
Line 82-83, were the drying times differing, or were they all conducted for 8 hours. Also, how was it decided, when to stop drying?
= All samples were conducted for 8 h. From previous study (Mizukami et al., 2006: moisture content measurement of tea leaves by electrical impedance and capacitance ) showed that moisture content (13%) in tea leaves though drying processing. From preliminary study, found that drying time (for 8 h) had moisture content (11-12%) and aw (0.4-0.5). For dedication to design drying time (for 8 h) due to focus on moisture content after drying which related to the study of mizukami et al. (2006).
Line 84, how could the fresh coffee sample be pulverised to powder?
=The fresh coffee leaves sample was cut to small pieces, it is not powder. Because of the fresh leaves had high moisture content.
Line 85, which samples were designated, which names, authors need to be clearer.
= Replace “Samples were labeled as fresh-young, dried-young and dried-mature” with “Samples were labeled as fresh young leaves, fresh mature leaves, dried young leaves and dried mature leaves”
Best regards
Samuchaya Ngamsuk, Ph.D. candidate
National Pingtung University of Science and Technology
Department of Tropical Agriculture and International cooperation

Reviewer 2 Report
The papers presents interesting results on coffee leaves content. data could be useful regarding the use of coffee leaves as antioxidant food (provided toxicity has been checked). There are some remarks for improvement:
-some typos correction and english polishing is needed: eg page 1 line 44: chlorogenic acid, page 4 line 179 Viscum album, page 6 line 282: compared; page 7 Figure 1 legend: dried young; check also figure 3 legend...
-one major remarks if the final expression of concentration. It is supposed to be mg/g of dried leaves. Was this also performed on fresh leaves taking into account the amount of water in fresh leaves ?
-data in Table 2 under DY40 do not match with same data in Table 3 under DY40: why ?
Other remarks:
-maturity leaves: quite vague ; please give better specification of mature leave
-in coffee leaves there is also a significant amount of cafestol, kawheol,actractylosides and derivatives ...
-method of leaves drying is not well described (time duration etc...)
-line 355: influence of temperature must also be mentioned
some literature devoted to the assay of coffee leaves is missing eg:
C.Mees et al in Talanta 177(2018)4-11
R.Rodriguez Gomez et al in Antioxidants 7(2018)143
F.Souard et al, Food Chem.245(2018)603-12
Author Response
Dear reviewer2
I revised the manuscript just a little bit (similar your suggestion) and add file answer
Answer
1.One major remarks if the final expression of concentration. It is supposed to be mg/g of dried leaves. Was this also performed on fresh leaves taking into account the amount of water in fresh leaves?
= Yes, it is supposed to be mg/g dried leaves. For fresh leaves, it was considered account the amount of water in fresh leaves.
2.Data in Table 2 Under DY40 do not match with same data in Table 3 Under DY 40:why?
= Regarding to Table 2, we investigated identification and quantification of catechin or epicatechin, mangiferin or isomangiferin, procyanidin B, caffeoylquinic acids (CQA), caffeine, quercetin-3-O-glucoside, procyanidin C, rutin, 3,4-dicaffeoylquinic acid from fresh leaves and dried leaves. After investigation, the result showed that dried young leaves had the highest phenolic compounds. Therefore, dried young leaves were used in next experiment. For table 3 starting with collecting, washing, sun-drying, rolling and tray dried leaves. The experiment was designed as varying of different temperature (30,40 and 50°C) for 8 hr and following extraction of dried young (different temperature) to investigate identification and quantification of of catechin or epicatechin, mangiferin or isomangiferin, procyanidin B, caffeoylquinic acids (CQA), caffeine, quercetin-3-O-glucoside, procyanidin C, rutin, 3,4-dicaffeoylquinic acid from dried leaves at different temperature.
Maturity leaves: quite vague: please give better specification of mature leave
= The difference between young leaves and mature leaves are position of leaves. Young leaves are on the apex whereas mature leaves are the first node as below the apex to second node.
In coffee leaves there is also a significant amount of cafestol, kawheol, actratylosides and derivatives….
= Yes it has several phenolic compounds. But this experiment can not detected some phenolic compounds due to using of different column, instrument, extracting solvent and processing when comparison to previous studies that have effect to the results.
method of leaves drying is not well described (time duration etc)
= Fresh coffee leaves (young and mature) were plucked and washed by hand. The coffee leaves were sun dried and rolled. 1 batch of the rolled coffee was tray dried for 8 h at 40°C to analyze the effect of drying process, and 3 other batches were tray-dried at different temperatures (30, 40 and 50°C for 8h), respectively,
line 355: influence of temperature must also be mentioned
= The different stages and different temperature caused significant differences in the antioxidant activity, total phenolic content and total procyanidins
Best regards
Samuchaya Ngasmuk
National Pingtung University of Science and Technology
Department of Tropical Agriculture and International cooperation

Round 2
Reviewer 2 Report
The revised form is not satisfactory:
-Authors did not provide in their MS informations regarding the origin of differences in Tables 2 and 3: for example in Table 2 DY40 mg/g for Mangifering is 49.83 and in Table 3 DY 40/mg for Mangiferin is 86.09 (double !!)
-Table 2 legend is NOT in accordance with content of the Table DY40 , DM40 .. ???
-Literature survey is not completed as suggested by referee
-Authors MUST provide correction in their MS and not only reply to the ref...
Author Response
Dear reviewer
I revised the manuscript just a little bit (like your suggestion) and add file answer
Best regards
Samuchaya Ngamsuk, Ph.D. candidate
National Pingtung University of Science and Technology
Department of Tropical Agriculture and International Cooperation
